# A Comparative Study on the Effects of the Lysine Reagent Pyridoxal 5-Phosphate and Some Thiol Reagents in Opening the Tl^+^-Induced Mitochondrial Permeability Transition Pore

**DOI:** 10.3390/ijms24032460

**Published:** 2023-01-27

**Authors:** Sergey M. Korotkov, Artemy V. Novozhilov

**Affiliations:** Sechenov Institute of Evolutionary Physiology and Biochemistry, Russian Academy of Sciences, Thorez pr. 44, 194223 St. Petersburg, Russia

**Keywords:** Tl^+^, Ca^2+^, pyridoxal 5-phosphate, thiol reagents, mitochondrial permeability transition pore

## Abstract

Lysine residues are essential in regulating enzymatic activity and the spatial structure maintenance of mitochondrial proteins and functional complexes. The most important parts of the mitochondrial permeability transition pore are F1F0 ATPase, the adenine nucleotide translocase (ANT), and the inorganic phosphate cotransporter. The ANT conformation play a significant role in the Tl^+^-induced MPTP opening in the inner membrane of calcium-loaded rat liver mitochondria. The present study tests the effects of a lysine reagent, pyridoxal 5-phosphate (PLP), and thiol reagents (phenylarsine oxide, tert-butylhydroperoxide, eosin-5-maleimide, and mersalyl) to induce the MPTP opening that was accompanied by increased swelling, membrane potential decline, and decreased respiration in 3 and 3U_DNP_ (2,4-dinitrophenol uncoupled) states. This pore opening was more noticeable in increasing the concentration of PLP and thiol reagents. However, more significant concentrations of PLP were required to induce the above effects comparable to those of these thiol reagents. This study suggests that the Tl^+^-induced MPTP opening can be associated not only with the state of functionally active cysteines of the pore parts, but may be due to a change in the state of the corresponding lysines forming the pore structure.

## 1. Introduction

Recent studies have found that the main structural parts of the mitochondrial permeability transition pore (MPTP) are very likely to be the Ca^2+^-modified ANT and F1F0-ATPase [1,2,3]. We previously showed that the interaction of Tl^+^ ions with calcium-loaded rat liver and rat heart mitochondria resulted in Tl^+^-induced mitochondrial permeability pore (MPTP) opening in the inner membrane [4,5]. This pore opening was accompanied by massive mitochondrial swelling, a marked decrease in mitochondrial respiration in states 3 and 3U_DNP_, and the inner membrane potential (ΔΨ_mito_) decline. We showed that the Tl^+^-induced MPTP opening became more palpable in experiments with thiol reagents (phenylarsine oxide (PAO), 4,4’-diisothiocyanostilbene-2, 2’-disulfonate (DIDS), high mersalyl (MSL), high n-ethylmaleimide (NEM)) and thiol oxidants (tert-butylhydroperoxide (*t*BHP), diamide (Diam)) [6,7,8]. On the contrary, the decrease in this opening has occurred in the presence of eosin-5-maleimide (EMA) and low MSL, and it was especially noticeable in experiments with bongkrekic acid (BKA), ADP, and low NEM [6,9].

It was previously shown that the MPTP opening becomes more probable if ANT is in the c-conformation, and, conversely, it visibly decreases when this exchanger is stabilized in the m-conformation [1,3,10,11,12]. So, our results above [5,6,7,8,9] are because the former reagents (PAO, DIDS, high MSL, high NEM, *t*BHP, and Diam) stabilize the ANT c-conformation, and the latter ones (ADP, EMA, BKA, low NEM, and low MSL) favor this exchanger in the m-conformation [10]. It is well known that the ANT cysteines play an essential role in maintaining this exchanger conformation. One of the most critical cysteines is Cys^159^ which is faced toward the matrix side [10]. If the former reagents interact with the ANT Cys^159^, the translocase c-conformation is fixed [10,12,13,14]. As a result, the affinity of calcium-binding sites to Ca^2+^ increases, but it falls to ADP [10,13,14,15]. EMA weakly penetrating IMM reacts with the ANT Cys^159^ with a markedly lower affinity [10,11,14,16]. If the reaction has a place with Cys^47^ in experiments with low NEM and low EMA, then stabilization of the m-conformation is observed [10,14,15]. Moreover, the histidine protonation and chemical modification of arginines are known to cause the closure of the mitochondrial permeability transition pore [17,18].

Lysine and cysteine residues are essential for maintaining many mitochondrial enzymes’ catalytic activity and optimal structure. Acetylated lysine residues contain 20% of mitochondrial proteins, including the F1 ATP synthase α, β, subunits that participate in the MPTP formation [1,19]. The covalent modification of lysine residues possibly affects the MPTP open probability [20]. Many mitochondrial enzymes (ligases, aminotransferases, synthases, hydroxymethyltransferases, and desulfurases) use PLP as their cofactor [21,22]. The reaction of PLP with protein is accompanied by the forming of a Schiff base [23]. In this regard, investigations often use PLP, which is known to be a lysine-selective reagent and universal mitochondrial transporter inhibitor [24,25,26,27,28,29]. Experiments with isolated mitochondria and omnifarious proteoliposomes showed that PLP inhibited or notably affected oxoglutarate/fumarate and malate/phosphate exchange, aspartate/glutamate, glutamine, and citrate (tricarboxylate) carriers as well as the transport of oxoglutarate, malonate, coenzyme A, inorganic phosphate, uridine-5’-diphospho-N-acetylglucosamine, and ascorbate [25,28,29,30,31,32,33,34,35,36,37,38,39,40,41]. On the other hand, PLP inhibited the proton-translocating NAD(P)^+^ transhydrogenase and pyruvate and malate dehydrogenases [42,43,44,45]. Both lysine and cysteine residues play an essential role in maintaining the ANT optimal structure [24]. If the ANT is in the m-state, then the matrix faced Lys^42^ and Lys^48^ are more susceptible to PLP than with the c-state [24]. MPTP opening was previously shown to have been stimulated by the ANT inhibitors (carboxyatractyloside, palmitoyl coenzyme A, and membrane-impermeant PLP) [2,46]. The ANT c-conformation was supposed to be stabilized due to the oxidation of sulfhydryl groups associated with the oxidation of mitochondrial pyridine nucleotides [47]. Currently, there are no studies on the role of the functional state of mitochondrial lysine residues in Tl^+^ toxic effects on mitochondria. Our goal was to conduct comparative studies on the combined impact of Tl+ with PLP or thiol reagents on the Tl^+^-induced MPTP opening in the inner membrane of Ca^2+^-loaded rat liver mitochondria. We investigated oxygen consumption rates in 4, 3, and 3U_DNP_ (2,4-dinitrophenol uncoupled) states, as well as swelling and ΔΨ_mito_ decline in medium containing TlNO_3_, KNO_3_, succinate, rotenone, and MPTP inhibitors (ADP, cyclosporine A (CsA), NEM) as well as Ca^2+^, PLP, and thiol reagents (where indicated).

## 2. Results

### 2.1. Effects of PLP and Thiol Reagents on the Tl^+^–Induced Swelling of Succinate-Energized Rat Liver Mitochondria

The swelling of succinate-energized mitochondria was slightly increased in the presence of 2–6 mM PLP (Figure 1A). At once, calcium-loaded rat liver mitochondria swelled a little more with 2–6 mM PLP in comparison to the control experiments (Figure 1A). ADP inhibited this swelling with minimal effect at 6 mM PLP (Figure 1B). The MPTP inhibitors significantly prevented the swelling of calcium-loaded RLM in the presence of 6 mM PLP (Figure 1C). This swelling decreased in the series CsA or PLP alone > control > ADP or NEM > CsA + NEM > ADP + CsA or ADP + NEM > “0” (free of Ca2+ and additions) (Figure 1C). CsA is interested in has not inhibited this swelling even in similar experiments with 2 mM PLP (not shown here). The swelling of succinate-energized mitochondria in experiments with thiol reagents and PLP increased in the series EMA < “0” (free of additions) < PLP or NEM(1) < NEM(2) < high MSL < *t*BHP < PAO (Figure 1D). A minor swelling increase (Figure 1E) was found in similar experiments with calcium-loaded mitochondria in the presence of EMA, PLP, MSL, and *t*BHP. Herewith, the similar effect of PAO was maximal. 

### 2.2. Effects of Tl^+^ and Thiol-Modifying Agents on Respiration and ΔΨ_mito_ of Succinate-Energized Rat Liver Mitochondria

One can see in Figure 2A that PLP partly inhibited states 3 and 3U_DNP_ respiration; however, the PLP effect on DNP-uncoupled respiration was not so pronounced. As for other thiol reagents, their inhibitory effect on RCR (Figure 2B) and state 3 respiration (Figure 2C) [6,9] increased in the series *t*BHP < EMA < PLP or MSL < PAO. DNP-uncoupled respiration [6,7,9] and RCR_DNP_ (Figure 2D) decreased similarly in experiments with these reagents. However, the inhibitory effect of PAO on RCR_DNP_ (Figure 2D) was more visible due to the PAO-induced state 4 increase [6]. The decrease in state 3U_DNP_ respiration and RCR_DNP_ in calcium-loaded RLM (control) was visibly greater in increasing PLP concentration in the medium containing TlNO_3_ and 2–6 mM PLP (Figs. 3A and B). The mitochondrial response to DNP added into the medium was preserved in this case. Thiol reagents (PAO and *t*BHP), similar to PLP and unlike EMA or MSL, contributed to a further decrease in both 3U_DNP_ respiration and RCR_DNP_ in experiments with calcium-loaded mitochondria (Figure 3A,B) [6,7,9]. The MPTP inhibitors (ADP with CsA or NEM alone) noticeably prevented a Ca^2+^-induced decrease in state 3U_DNP_ respiration or RCR_DNP_ in these experiments with/without PLP (Figure 3A,C). The Ca^2+^-induced respiration decrease was markedly prevented by the MPTP inhibitors in the medium containing both PLP and *t*BHP or PAO (Figure 3B,C) [6,7,9]. On the other hand, the above mitochondrial parameters were affected by EMA or MSL, similar to the MPTP inhibitors used. The additive effect of MSL with EMA was observed in this case. The fluorescent dye, safranin O, was used to evaluate the inner mitochondrial membrane potential (ΔΨ_mito_). The addition of succinate into the medium led to the uptake of dye by succinate-energized mitochondria due to the inner membrane potential appearance. As a result, a decrease in the mitochondrial suspension fluorescence was observed. Calcium visibly decreased the membrane potential. An even greater decrease in the potential (Figure 4) was observed after adding calcium to the medium containing thiol reagents (PLP, EMA, MSL, *t*BHP, and PAO) [6,7,9]. However, this decrease was completely eliminated in the medium containing ADP and CsA.

## 3. Discussion

PLP was shown to penetrate slowly and passively into the mitochondrial matrix along a concentration gradient regardless of the presence of inhibitors and oxidative phosphorylation uncouplers [21,48]. The manganese transporter (Mtm1) transfers PLP into the matrix with micromolar affinity [49]. PLP and PAO were found to have produced increased swelling, decreased respiration in 3 and 3U_DNP_ states, and ΔΨ_mito_ decline in the RLM experiments (Figure 1, Figure 2 and Figure 4) [6]. However, the above PLP effects occurred at a much lower rate since PLP concentrations of three orders of magnitude more than PAO concentrations are required to produce an impact of similar magnitude. Other thiol reagents (*t*BHP, Diam, DIDS, MSL, and high MSL) similarly affected swelling and respiration in 3 and 3U_DNP_ states in experiments with calcium-free media (Figure 1, Figure 2 and Figure 4) [50]. The similarity of these effects may be because PLP can react not only with the ANT lysine residues but PLP can also interact with the ANT cysteine residues, similar to the reaction of PAO and above reagents with the cystein ones [10,11,16,24,28,29,50]. Low concentrations of PLP and thiol reagents were insufficient to inhibit visible RCR and state 3 respiration (Figure 2A–C) [6,7,8,9,50]. Wherein, state 3, not state 4, was markedly inhibited by these reagents at high concentrations. The state 3 inhibition increased in the series *t*BHP, Diam < PAO < EMA < PLP, FITC < high MSL, DIDS, high NEM (Figure 2C) [6,50]. RCR showed similar series, but PAO-induced RCR decrease was maximal due to the state 3 decline and state 4 increase [6].

State 3 respiration is known to depend on the activities of F_1_F_0_-ATP synthase, the adenine nucleotide translocase (ANT), and the mitochondrial phosphate cotransporter (PiC), which are necessary for the ATP synthesis carried out by mitochondria. PLP and DIDS can inhibit F_1_F_0_-ATP synthase, gastric H^+^/K^+^-ATPase, and tonoplast ATPase [6,51,52,53,54]. The ANT inhibition was observed in the presence of PLP [24,55], EMA [30], and FITC (discussed in [9]) as well as PAO, *t*BHP, and Diam (discussed in [6]). This inhibition resulted in the reaction of these reagents with the ANT cysteine residues. However, FITC does not interact with ANT cysteines, but inhibits ADP transport in mitochondria and reacts with the PiC cysteines [9,30]. The mitochondrial H^+^/Pi cotransporter activity was notably decreased in experiments with PLP [16,41], low MSL [7], EMA [56], and FITC [9]. State 3 respiration decrease may be due to inhibiting the enzymes involved in the oxidative phosphorylation processes and blocking the respiratory substrate transport into mitochondria. PLP and some thiol reagents (PAO, DIDS, FITC, and high MSL) inhibit a succinate transport into the matrix [6,7,9]. At the same time, state 3U_DNP_ respiration decrease [6,7,9] was insignificant in experiments with mitochondria, energized substrates of the first respiratory complex (glutamate with malate) in the presence of reagents (DIDS, MSL, EMA, and FITC). Thus, the succinate transport inhibition may explain the marked decrease in state 3U_DNP_ respiration and RCR_DNP_ values (Figure 2A,D) in experiments with the latter reagents. By comparing the effects of other reagents (PLP, *t*BHP, Diam, and EMA) on states 3 and 3U_DNP_ (Figure 2), it can be concluded that they have a weak impact on succinate transport.

Some used reagents (PAO, MSL, DIDS, *t*BHP, and Diam) showed increased swelling in experiments with Ca^2+^-loaded mitochondria (Figure 1) [9,50]. A similar effect of PLP was minor (Figure 1), whereas EMA and DIDS partially prevented this increase [6,9]. The Ca^2+^-induced decrease in state 3U_DNP_ respiration was more visible in experiments with reagents (PAO, *t*BHP, Diam, FITC, and PLP) in comparison to DIDS, MSL, and EMA that slowed down these effects (Figure 3) [9,50]. Similar results are associated with the involvement of ANT in the MPTP opening under the action of thiol reagents (PAO, MSL, and DIDS) and stress inducers (*t*BHP and Diam) on mitochondria due to the stabilization of this exchanger c-conformation because of their reaction with the ANT cysteines [6,50]. Opposite, DIDS and low MSL decreased the MPTP opening due to ANT conformational changes, altering the thiol group reactivity [6,7]. In addition, MSL or DIDS presence discovered that the state 3U_DNP_ respiration decrease was not so pronounced in calcium-loaded mitochondria injected in the medium with TlNO_3_ (Figure 3B) [50]. These results suggest that the interaction of these reagents with the PiC, not ANT thiol groups, plays a primary role. EMA, compared to NEM, showed a more significant affinity to ANT cysteine thiol groups, but EMA weak penetrates across the inner mitochondrial membrane [9,10,11,14,16]. EMA inhibited beef heart mitochondria swelling due to this reagent interacting with the cytoplasm-faced PiC essential thiol groups [9,56]. If the reaction has a place with Cys^47^ in experiments with low NEM and low EMA, then the m-conformation stabilization is observed [10,13,14]. The EMA block of the state 3U_DNP_ respiration decrease (Figure 3B) may be due to the EMA reaction with the ANT Cys^47^ that followed this exchanger m-conformation stabilization [10,14,15]. The PLP effects on calcium-loaded mitochondria are most likely associated with the interaction of PLP with the ANT cysteine residues [24].

The MPTP inhibitors (ADP, EMA, and low NEM) are known to fix ANT in the m-conformation and to prevent the increased swelling and ΔΨ_mito_ decline in experiments with Ca^2+^-loaded mitochondria in the presence of thiol reagents (PAO, *t*BHP, Diam, EMA, arsenite, or menadione) that react with the ANT Cys^159^ and Cys^256^ [10,11,13,14]. The ADP inhibition of Tl^+^-induced MPTP opening was less pronounced in experiments with PLP, PAO, and DIDS than with MSL, EMA, and FITC (Figure 1, Figure 3 and Figure 4) [6,7,9,51]. Such a difference in effects may result from the fact that the former actively interacts with ANT cysteines, while the latter did not reveal such a critical effect on the ANT structure [6,7,9,11,24,56]. Thus, this circumstance ultimately allowed ADP to inhibit the MPTP opening. Low NEM (50 µM) prevented the increase in swelling and the decrease in RCR_DNP_ and state 3U_DNP_ respiration in experiments with the used thiol reagents (Figure 1C and Figure 3C) [6,7,8,9,50]. The Tl^+^-induced MPTP inhibition was due to the NEM interaction with the above ANT cysteines in experiments with PAO, Diam, EMA, *t*BHP, and PLP [10,11,13,14]. However, FITC- and MSL-induced MPTP opening in TlNO_3_ media (Figure 1C and Figure 3C) was inhibited by low NEM due to the reaction of NEM with cytoplasm-faced PiC thiols [7,9,50]. CsA visibly inhibited the Tl^+^-induced MPTP opening in experiments with *t*BHP, EMA, FITC, and MSL, but not PLP (Figure 1C and Figure 3C) [6,7,9,50]. This result suggests that CyP-D is not involved in the MPTP opening in experiments with PLP. In this case, the increased swelling decreased respiration, and ΔΨmito decline in experiments with used thiol and lysine (PLP) reagents were noticeably leveled or completely disappeared in simultaneous presencing NEM with ADP or NEM with CsA in the experiments with the TlNO_3_ medium. The inhibitory effect of ADP with CsA was maximal (Figure 1C, Figure 3C and Figure 4) [6,7,9,50].

## 4. Materials and Methods

### 4.1. Animals

Male Wistar rats (250–300 g) were kept at 20–23 °C under a 12 h light/dark cycle with free access to water ad libitum and the standard rat diet. All treatment procedures of animals were performed according to the Animal Welfare act and the Institute Guide for Care and Use of Laboratory Animals (Protocol # 6/5/2022).

### 4.2. Chemicals

Sucrose, CaCl_2_, Mg(NO_3_)_2_, H_3_PO_4_, KNO_3_, TlNO_3_, and 2,4-dinitrophenol (DNP) were of analytical grade from Nevareactiv (St. Petersburg, Russia). Rotenone, oligomycin, pyridoxal-5-phosphate (PLP), PAO, *t*BHP, NEM, safranin O, tris-OH, ethylene glycol tetraacetic acid EGTA, ADP, CsA, and succinic acid were from Sigma (St. Louis, MO, USA). Sucrose as a 1 M solution was refined from cation traces on a column filled with a KU-2-8 resin from Azot (Kemerovo, Russia).

### 4.3. Mitochondrial Isolation

Rat liver mitochondria were isolated according to [9] in a buffer containing 250 mM sucrose, 3 mM Tris-HCl (pH 7.3), and 0.5 mM EGTA; subsequent mitochondrial sediment was washed out twice by resuspension-centrifugation in a medium containing 250 mM sucrose and 3 mM Tris-HCl (pH 7.3) and finally suspended in 1 mL of the last buffer. According to Bradford, the mitochondrial protein content assayed was within 50–60 mg/mL.

### 4.4. Swelling of Mitochondria

Mitochondrial swelling (Figure 1) was tested as a decrease in A_540_ at 20 °C using an SF-46 spectrophotometer (LOMO, St. Petersburg, Russia). Mitochondria (1.5 mg of protein/mL) were injected into a 1-cm cuvette with 1.5 mL of 400 mOsm medium containing 75 mM TlNO_3_, 125 mM KNO_3_, 5 mM Tris-NO_3_ (pH 7.3), 2 μM rotenone, and 1 μg/mL of oligomycin. PLP, PAO, *t*BHP, NEM, Ca^2+^, succinate, NEM, ADP, and CsA were added into the medium before or after mitochondria (see Figure 1 legend). The swelling, oxygen consumption rates, and ΔΨ_mito_ dissipation were investigated in 400 mOsm media to verify the comparability and consistency between data in different experiments.

### 4.5. Oxygen Consumption Assay

Mitochondrial respiration (oxygen consumption rate) was measured polarographically using an Expert-001 analyzer (Econix-Expert Ltd., Moscow, Russia) in a 1.3 mL closed thermostatic chamber with magnetic stirring at 26 °C. Mitochondria (1.5 mg of protein/mL) were added to 400 mOsm medium containing 25 mM TlNO_3_, 100 mM sucrose, 3 mM Mg(NO_3_)_2_, and 3 mM Tris-P_i_ (Figure 2) or 75 mM TlNO_3_ and 1 μg/mL of oligomycin (Figure 3) as well as 125 mM KNO_3_, 5 mM Tris-NO_3_ (pH 7.3), 5 mM succinate, and 2 μM rotenone. PLP, PAO, *t*BHP, EMA, MSL, Ca^2+^, NEM, ADP, and CsA were added in the medium before or after mitochondria (see Figure 2 and Figure 3 legend). ADP of 130 μM (Figure 2) and 30 μM DNP (Figure 2 and Figure 3) were correspondingly added into the medium after 2 min recording of state 4 to induce state 3 and 3U_DNP_ respiration. The respiratory control ratio (RCR_ADP_) was calculated as a ratio of state 3 to state 4 (Figure 2). The RCR_DNP_ was accordingly quantified as a ratio of DNP-uncoupled respiration to state 4 (Figure 2) or state 4_0_ respiration (Figure 3).

### 4.6. Mitochondrial Membrane Potential

The inner membrane potential (ΔΨ_mito_) induced by injection of 5 mM succinate into a medium was tested according to Waldmeier et al. [57]. The safranin O fluorescence intensity (arbitrary units) in the mitochondrial suspension was tested at 20 °C (Figure 4) using the microplate reader (CLARIOstar^®^ Plus, BMG LABTECH, Ortenberg, Germany) at 485/590 nm wavelength (excitation/emission). The mitochondria (0.5 mg of protein/mL) were added into cells containing 300 μL of the medium with 20 mM TlNO_3_, 125 mM KNO_3_, 110 mM sucrose, 5 mM Tris-NO_3_ (pH 7.3), 1 mM Tris-P_i_, 2 μM rotenone, 3 μM safranin O, and 1 μg/mL of oligomycin. ADP, CsA, PLP, EMA, PAO, *t*BHP, and MSL were injected into the medium before mitochondria (see Figure 4 legend). 5 mM succinate, 75 μM Ca^2+^, and 30 μM DNP was administrated into the medium after the mitochondria. The safranin O fluorescence change after the succinate injection was taken as 100% in control experiments free of the above additions (PLP, thiol reagents, ADP, CsA, and Ca^2+^). The other cases’ fluorescence values were calculated relative to this control. A parallel fourfold measurement for each 300 μl aliquot was made from three independent preparations.

### 4.7. Statistical Analysis

The statistical differences in results and corresponding *p*-values were evaluated using two population *t*-tests (Microcal Origin, Version 6.0, Microcal Software). These differences are presented as a percent of the average (*p* < 0.05) from one of three independent experiments (Figure 1, Figure 2, Figure 3 and Figure 4). A more detailed statistical analysis is in supplementary materialsAppendix A.

## 5. Conclusions

Mitochondrial proteins contain up to 20% acetylated lysine residues, including F_1_F_0_-ATP synthase structural parts [19]. The covalent modification of lysine residues is believed to affect the MPTP open probability [13,20]. The MPTP opening resulted in a fluorescamine reaction with the ANT lysine residues that induced efflux of accumulated Ca^2+^, ΔΨ_mito_ decline, and mitochondrial swelling [58]. Experiments with BKA- and CAT-treated mitochondria showed that one or more lysine residues could be involved in CAT and BKA binding with the ANT [24,55]. Lys^401^ of bovine heart mitochondrial F1F0-ATP synthase was modified explicitly with 4-chloro-7-nitrobenzofurazan [59]. Pyridoxal 5’-diphospho-5’-adenosine was shown to bind to the isolated alpha-subunit from *E. coli* F_1_-ATP synthase [60]. FITC reacts with cysteine and the PiC lysine residues. It quickly penetrates the mitochondrial matrix, binds to α and γ subunits of the F1-ATP synthase, and localizes in the inner membrane lipid part [9,30]. FITC binds the PiC Lys residue more efficiently than PLP [30]. It can be seen that lysine residues are essential for the functioning of the MPTPs most important parts, namely F1F0-ATP synthase, ANT, and Pi cotransporter. An analysis of the study results allows us to conclude that the Tl^+^-induced MPTP opening is associated not only with functionally active cysteines of the latter three parts, but may also be related to activities of the corresponding lysines that form the pore structure parts.

## Figures and Tables

**Figure 1 ijms-24-02460-f001:**
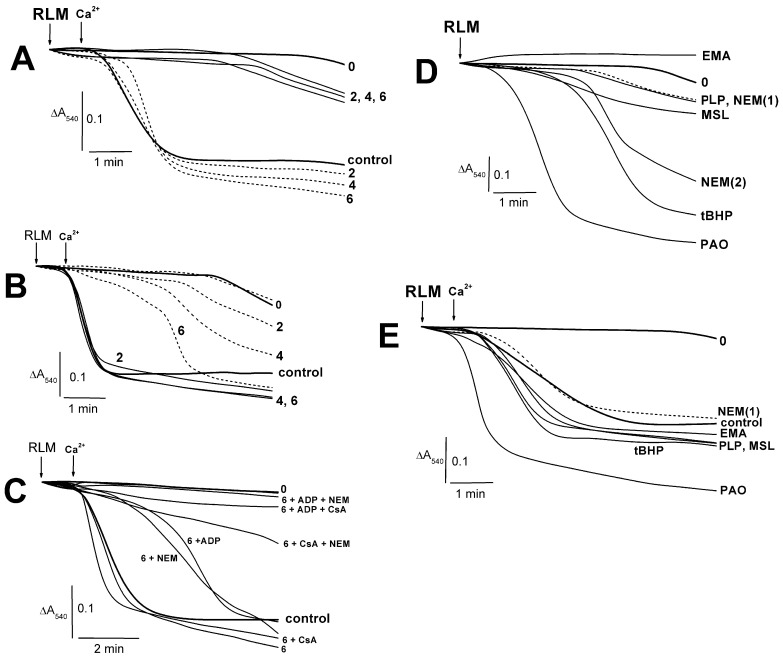
Effects of Ca^2+^, pyridoxal 5-phosphate, and thiol reagents on the Tl^+^-induced swelling of succinate-energized rat liver mitochondria. Mitochondria (1.5 mg of protein per mL) were injected into the medium containing 75 mM TlNO_3_, 125 mM KNO_3_, 100 μM Ca^2+^ (where indicated), and 5 mM Tris-succinate (pH 7.3). In addition, the medium was supplemented with 500 μM ADP ((**B**) short dash traces). The following reagents were added into the medium before mitochondria ((**C**–**E**), indicated to the right of traces) 500 μM ADP (ADP), 1 μM CsA (CsA), 50 μM NEM (NEM(1)), 500 μM NEM (NEM(2)), 10 μM EMA (EMA), 20 μM MSL (MSL), 6 mM PLP (PLP), 100 μM *t*BHP (*t*BHP), and 5 μM PAO (PAO). Additions of mitochondria (RLM) and 100 μM Ca^2+^ (Ca^2+^) are shown by arrows. Experiments free of Ca^2+^ (solid traces) or ones with 100 μM Ca^2+^ (short dash traces) showed on panel **A**. The numbers to the right of the traces (**A**–**C**) show concentrations (μM) of PLP, which was added into the medium before mitochondria. The bold traces show experiments free of Ca^2+^ and PLP (0) or ones with Ca^2+^ alone (control). Representative traces from one of three independent experiments are presented.

**Figure 2 ijms-24-02460-f002:**
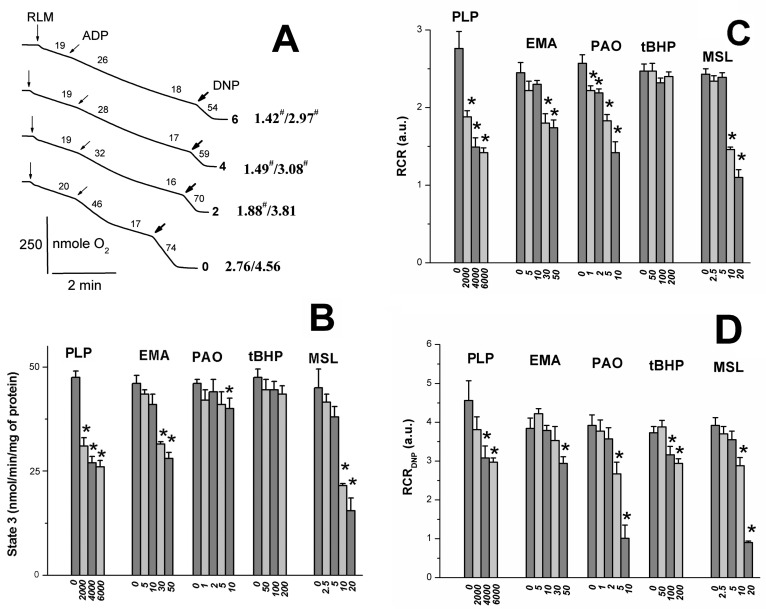
Effects of pyridoxal 5-phosphate and thiol reagents on oxygen consumption rates of rat liver mitochondria. Mitochondria (1.5 mg/mL of protein) were added into a medium containing 25 mM TlNO_3_, 125 mM KNO_3_, 100 mM sucrose, 5 mM Tris-NO_3_ (pH 7.3), 3 mM Tris-P_i_, 3 mM Mg(NO_3_)_2_, 5 mM succinate, and 2 μM rotenone. Additions of mitochondria (RLM), 130 μM ADP (ADP), and 30 μM DNP (DNP) are correspondingly shown by vertical arrows, inclined arrows, and bold arrows. Oxygen consumption rates (nmole O_2_ min/mg of protein) are presented as numbers placed above experimental traces. The numbers to the right of the traces (**A**) show PLP (mM) concentrations. Numbers (**A**) in bold with the slash between them to the right of the traces, respectively, show values of the RCR_ADP_ and the RCR_DNP_ (see the Section 4). The hash to the right of the latter numbers indicates that the difference between appropriate values of the RCR_ADP_ and the RCR_DNP_ is statistically insignificant to the values found in experiments free of PLP. Representative traces from one of three independent experiments are presented. The ordinate (**B**) marks oxygen consumption rates (nmole O_2_ min/mg of protein) in state 3. The ordinates (**C**,**D**) show RCR and RCR_DNP_ values. Numbers below the abscissa (**B**–**D**) indicate the concentration (μM) of thiol reagents (PLP, EMA, MSL, PAO, and *t*BHP). * shows significant differences from the control (“0” under the abscissa) experiments free of thiol reagents (*p* < 0.05).

**Figure 3 ijms-24-02460-f003:**
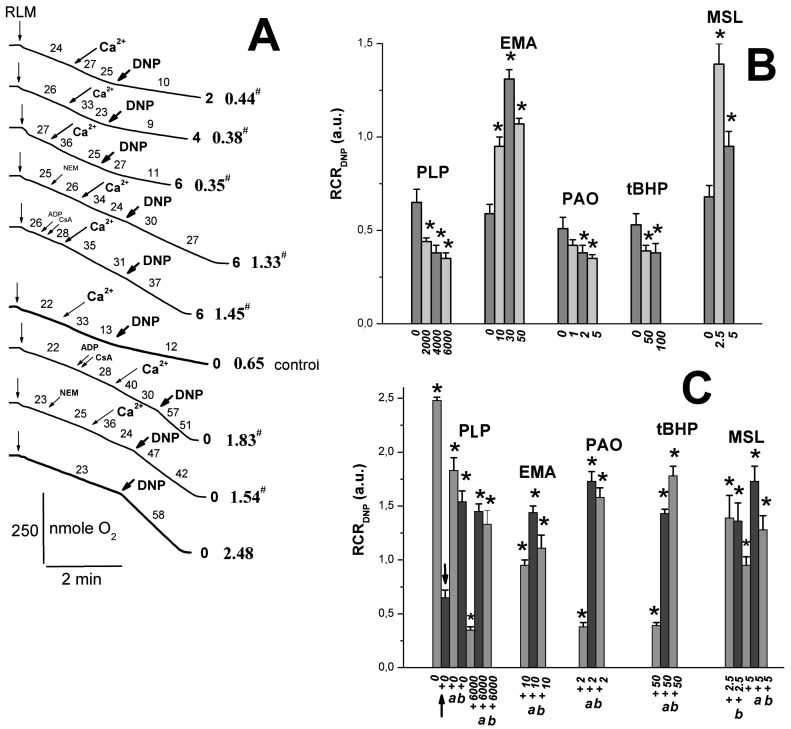
Effects of Ca^2+^, pyridoxal 5-phosphate, and thiol reagents on oxygen consumption rates of rat liver mitochondria. Mitochondria (1.5 mg/mL of protein) were added into a medium containing 75 mM TlNO_3_, 125 mM KNO_3_, 5 mM Tris-NO_3_ (pH 7.3), 5 mM succinate, 2 μM rotenone, and 1 μg/mL of oligomycin. Additions of mitochondria (RLM), 100 μM Ca^2+^ (Ca^2+^), and 30 μM DNP (DNP) are correspondingly shown by the vertical arrows, inclined long arrows, inclined long bold arrows, and bold arrows. Inclined arrows indicate additions of 500 μM ADP (ADP), 1 μM CsA (CsA), and 50 μM NEM (NEM). The numbers (**A**) to the right of the traces show concentrations of PLP (mM). Oxygen consumption rates (nmole O_2_ min/mg of protein) are presented as numbers placed above experimental traces. Numbers (**A**) to the right of the traces in bold show the value of the RCR_DNP_. Experiments free of additions (**A**) are indicated to the right of the traces: free of Ca^2+^ and PLP (0) and 100 μM Ca^2+^ alone (control). Representative traces from one of three independent experiments are presented. The ordinates (**B**,**C**) show RCR_DNP_ values. Designations (with axes and concentration reagents) on panels (**B**) and (**C**) are the same as in Figure 2 (panels (**C**,**D**)). The plus sign (**C**) indicates experiments with 100 μM Ca^2+^, and Latin letters (a,b) indicate experiments correspondingly with 500 μM ADP and 1 μM CsA (a) or 50 μM NEM alone (b), added after mitochondria as in panel (**A**). * shows significant differences (*p* < 0.05) from the control (“0” under the abscissa) experiments free of thiol reagents (**B**) or Ca^2+^ alone (**C**, column enclosed between arrows).

**Figure 4 ijms-24-02460-f004:**
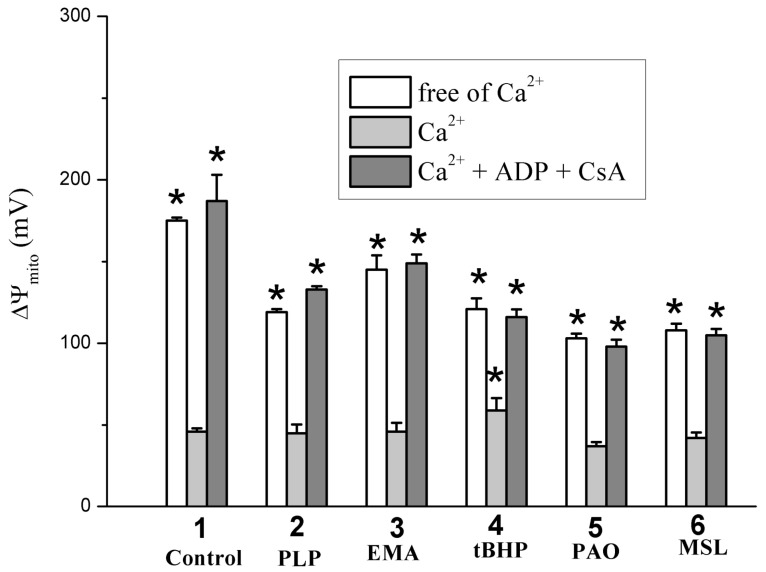
The influence of Ca^2+^, pyridoxal 5-phosphate, and thiol reagents on the inner membrane potential (ΔΨ_mito_). Mitochondria (0.5 mg/mL of protein) were injected into the medium containing 20 mM TlNO_3_, 125 mM KNO_3_, 110 mM sucrose, 5 mM Tris-NO_3_ (pH 7.3), 1 mM Tris-P_i_, 2 μM rotenone, 3 μM safranin O, and 1 μg/mL of oligomycin. The ordinate shows ΔΨ_mito_ change (mV) calculated from the safranine O fluorescence intensity (see Section 4). Numerals on the abscissa indicate the used reagents: control experiments (1), 6 mM PLP (2), 10 μM EMA (3), 100 μM *t*BHP (4), and 2 μM PAO (5). Besides, 500 μM ADP and 1 μM CsA (where indicated) were correspondingly added into the medium before mitochondria and Ca^2+^. Next, 75 μM Ca^2+^ was injected into the medium after mitochondria. * shows significant differences from the control experiments with Ca^2+^ and free of the reagents (*p* < 0.05).

## Data Availability

The data that support the findings of this study are available from the corresponding authors, [S.M.K.], upon reasonable request.

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
