# Peer review of "A Comparative Study on the Effects of the Lysine Reagent Pyridoxal 5-Phosphate and Some Thiol Reagents in Opening the Tl+-Induced Mitochondrial Permeability Transition Pore"

_ijms, 2023, doi:10.3390/ijms24032460_

Round 1

Reviewer 1 Report

Authors studied the effects of a lysine reagent, pyridoxal 17 5-phosphate (PLP), and thiol reagents (phenylarsine oxide, tert-butylhydroperoxide, eosin-5-malei-18 mide, mersalyl) to induce the MPTP opening that accompanied by the increased swelling, the mem-19 brane potential decline, and the decreased respiration in 3 and 3UDNP (2,4-dinitrophenol-uncou-20 pled) states. The present manuscript presents a novel work and to improve the quality of the manuscript I suggest  some minor comments  kindly authors can  display them  in the attachment pdf file. 

Author Response

Assigned Editor

Carlie Chen

IJMS Editorial Office

Dear Dr. Chen

There I present my replies to the two reviewer' comments on our manuscript "Comparative study of effects of the lysine reagent pyridoxal 5-phosphate and some thiol reagents in opening the Tl+-induced mitochondrial permeability transition pore". Despite a certain share of criticism, we are very grateful to these reviewers because of these remarks were very useful to improve our manuscript. Across new version our manuscript my corrections are highlighted in green (for 1st reviewer) and yellow (for 2nd revievwer). Please find our point by point response to each of the comments.

Sergey Korotkov, PhD in Biochemistry and MS in Chemistry

January 24, 2023

St. Petersburg

My Responses to Reviewer 1.

1.

Title is too long, and too confusing for reader, i suggest if you can conceder compress the main idea in short title”.

The manuscript title was condensed to “Comparative study effects of the lysine reagent pyridoxal 5-phosphate and some thiol reagents in opening the Tl+-induced mitochondrial permeability transition pore”. The former name was “Comparative study effects of the lysine reagent pyridoxal 5-phosphate and some thiol reagents in opening the Tl+-induced permeability transition pore in experiments in vitro with calcium-loaded rat liver mitochondria”.

2.

I suggest to insert graphical abstract to illustrate the results”.

The graphical abstract inserted on Page 16.

3.

“The authors can modify the quality of Figures (2, 3 and 4)”.

I inserted the quality Figures (1-4) into the manuscript text.

4.

“1- How many rats used”?  

For isolation of the rat liver mitochondrial preparation, one rat was used each time.

2- Is there an Ethical certificate number for this experiment?  

The ethical certificate number is on Page 8 (4.1. Animals). The Ethical certificate inserted on Page 17.

3- The authors could illustrate the experimental design it will be helpfully for the reader. This section needs to be described briefly. 

The experiment scheme is well illustrated in the methods, so there is no need to briefly describe this section.

Reviewer 2 Report

Dear Authors, please correct some errors: title text - Comparative study of ...; P.3, line 99 - text should be changed,  better use" similar" instead of "like"; P.5, lines 157-158 - italics should be shanged to normal text; ; P.7, line 198 - by these reagents ...;  P.8, line 278 - what producer s are for sucrose and TROS-HCl; ; P.10, line 355 - use italics for E.coli; P.11, line 373 - font size should be changed as necessary; ; Acknowledgements - line 387, the authors are gratefull  for ...estimation of the oxygen ...; Abbreviations - line 389 - 2,4-Dinitrophenol -uncoupled respiration; NEM -n-Ethylmaleimide...; PLP - Pyridpxal... References, N.26 - use not capital letters in the article title, except first name.

Author Response

Assigned Editor

Carlie Chen

IJMS Editorial Office

Dear Dr. Chen

There I present my replies to the two reviewer' comments on our manuscript "Comparative study effects of the lysine reagent pyridoxal 5-phosphate and some thiol reagents in opening the Tl+-induced mitochondrial permeability transition pore". Despite a certain share of criticism, we are very grateful to these reviewers because of these remarks were very useful to improve our manuscript. Across new version our manuscript my corrections are highlighted in green (for 1st reviewer) and yellow (for 2nd revievwer). Please find our point by point response to each of the comments.

Sergey Korotkov, PhD in Biochemistry and MS in Chemistry

January 24, 2023

St. Petersburg

My Responses to Reviewer 2.

 “Dear Authors, please correct some errors: title text - Comparative study of ...;” – corrected.

P.3, line 99 - text should be changed,  better use" similar" instead of "like";” - replaced.

P.5, lines 157-158 - italics should be changed to normal text;” – corrected.

P.7, line 198 - by these reagents ...;”  – corrected.

P.8, line 278 - what producers are for sucrose and TROS-HCl;” -  

Sucrose (from Nevareactiv (St. Petersburg, Russia)) is listed as a reagent (4.2. Chemicals). We used Tris-OH base (Sigma). Tris-HCl was resulted in titrating a medium by HCl.  

P.10, line 355 - use italics for E. coli;” – corrected.

P.11, line 373 - font size should be changed as necessary;” – corrected.

Acknowledgements - line 387, the authors are gratefull  for ...estimation of the oxygen ...;” – corrected.

Abbreviations - line 389 –

2,4-Dinitrophenol-uncoupled respiration;” – corrected to “2,4-Dinitrophenol uncoupled”.

NEM -n-Ethylmaleimide...;” - corrected.

PLP - Pyridpxal.” - corrected.

References, N.26 - use not capital letters in the article title, except first name”. - - corrected.
